# Short-Term Effects of Different Types of Anti-Glaucoma Eyedrop on the Sclero-Conjunctival Vasculature Assessed Using Anterior Segment OCTA in Normal Human Eyes: A Pilot Study

**DOI:** 10.3390/jcm9124016

**Published:** 2020-12-11

**Authors:** Tadamichi Akagi, Yoko Okamoto, Takanori Kameda, Kenji Suda, Hideo Nakanishi, Masahiro Miyake, Hanako Ohashi Ikeda, Tatsuya Yamada, Shin Kadomoto, Akihito Uji, Akitaka Tsujikawa

**Affiliations:** Department of Ophthalmology and Visual Sciences, Kyoto University Graduate School of Medicine, Kyoto 6068507, Japan; yoko616@kuhp.kyoto-u.ac.jp (Y.O.); kame@kuhp.kyoto-u.ac.jp (T.K.); qwm04525@kuhp.kyoto-u.ac.jp (K.S.); hideon@kuhp.kyoto-u.ac.jp (H.N.); miyakem@kuhp.kyoto-u.ac.jp (M.M.); hanakoi@kuhp.kyoto-u.ac.jp (H.O.I.); yamata2@kuhp.kyoto-u.ac.jp (T.Y.); kadomoto@kuhp.kyoto-u.ac.jp (S.K.); akihito1@kuhp.kyoto-u.ac.jp (A.U.); tujikawa@kuhp.kyoto-u.ac.jp (A.T.)

**Keywords:** anterior segment optical coherence tomography angiography, anti-glaucoma eyedrop, aqueous humor outflow pathway, intraocular pressure reduction

## Abstract

Background: To investigate the short-term effects of different types of anti-glaucoma eyedrop on sclero-conjunctival vasculatures and their associations with intraocular pressure (IOP) reduction. Methods: This was a prospective study including 20 healthy subjects. A single instillation of ripasudil or bimatoprost was introduced into the right eyes of the participants. The superficial (conjunctival) and deep (intrascleral) vasculatures of the corneal limbus using anterior-segment optical coherence tomography angiography (OCTA) and IOP were examined in both eyes at baseline and 15 min and 2 h after instillation. Results: In the ripasudil group, the vessel density (VD) (median) at baseline (deep, 13.1%; superficial, 28.5%) significantly increased in both layers at 15 min (deep, 19.9%; superficial, 37.3%) and the deep layer at 2 h (deep, 14.8%; superficial, 31.6%). In the bimatoprost group, the superficial VD significantly changed over time, but the deep VD did not. The greater effect of ripasudil on IOP reduction was significantly associated with a lower baseline VD in the deep layer (at 15 min, *p* = 0.004; at 2 h, *p* = 0.018). Conclusions: Differences in the timing, depth, and extent of the effects on vasculature after instillations, could be detected using OCTA. The IOP-lowering effects of ripasudil might be associated with the deep vasculature.

## 1. Introduction

Reduction of intraocular pressure (IOP) is the primary treatment strategy to prevent glaucoma progression. Conjunctival hyperemia is frequently observed as an adverse effect of various anti-glaucoma eye drops, and it disturbs the adherence to glaucoma therapy [1]. Conjunctival hyperemia is usually evaluated by standard grading scales [2,3,4,5]; however, objective and precise evaluation of the extent of conjunctival hyperemia is difficult, and deeper vasculatures such as the episcleral and intrascleral vasculatures show limited detectability on the basis of appearance. Nevertheless, these vasculatures, which include the episcleral veins (ESVs), are known as the distal portion of the conventional aqueous humor outflow (AHO) pathway; therefore, the episcleral vasculature, in particular, may have an important role in controlling IOP [6].

Ripasudil hydrochloride hydrate 0.4% (Glanatec; Kowa Company, Ltd., Nagoya, Japan) is a Rho-associated protein kinase (ROCK) inhibitor that effectively lowers IOP by increasing AHO by reducing resistance in the trabecular meshwork (TM) [7,8]. The conjunctival hyperemia induced by ripasudil instillation reportedly peaks at approximately 5–15 min after ripasudil administration and generally resolves within 2 h [4]. Bimatoprost 0.03% (Lumigan ophthalmic solution 0.03%; Allergan, Irvine, CA, USA) is a prostaglandin analog (PGA), which are the most widely used first-line drugs for the treatment of glaucoma [9]. Conjunctival hyperemia is the most common adverse effect of PGAs, and bimatoprost was reported to show the strongest extent of hyperemia among commercially available PGAs [2,10].

Anterior segment (AS)-optical coherence tomography angiography (OCTA) has enabled non-invasive visualization of the microvasculature of the corneo-scleral limbus [11]. We had previously reported that the intrascleral AS-OCTA flow images at least partly represented the post-TM AHO routes [12]. Furthermore, the superficial vessel density (VD) is significantly associated with the use of PGAs, and deep VD was significantly associated with IOP in treated eyes with glaucoma [13]. However, the effects of eye drops on AS-OCTA images have not been clarified. In this study, we prospectively investigated the time-dependent changes in superficial and deep AS-OCTA flow images and the associations between changes in IOP and VD by using a swept-source optical coherence tomography (OCT) system after a single instillation of ripasudil or bimatoprost.

## 2. Materials and Methods

### 2.1. Study Design and Participants

This prospective longitudinal study adhered to the tenets of the Declaration of Helsinki, was approved by the institutional review board and ethics committee of the Kyoto University Graduate School of Medicine (Kyoto, Japan), and was registered with the University Hospital Medical Information Network (UMIN) Clinical Trials Registry of Japan (UMIN000033375). Written informed consent was obtained from all participants. This study included 20 normal healthy volunteers (10 for ripasudil instillation and 10 for bimatoprost instillation) with no history of ocular or systemic disease. The ripasudil instillation study was performed between 1 October 2018, and 30 January 2019, and the bimatoprost instillation study was performed between 1 December 2019, and 28 February 2020.

### 2.2. Study Protocol for Eye Drop Instillation

All participants underwent an ophthalmic examination, which included slit-lamp examination and measurement of axial length (IOLMaster 500; Carl Zeiss Meditec, Dublin, CA, USA), central corneal thickness (SP-3000; Tomey, Tokyo, Japan), and IOP (Icare PRO Rebound Tonometer; Tiolat Oy, Helsinki, Finland), AS-OCTA examinations, and slit-lamp photography on both eyes at baseline. The IOP measurement method using the rebound tonometer in this study offered the advantage of allowing measurements to be obtained without changing the participant’s position and with minimal effects on the ocular surface; moreover, these measurements were reported to be highly correlated with those obtained by the Goldmann applanation tonometer, the gold standard for IOP measurement [14]. Immediately after baseline examination, a single instillation of the eyedrop (ripasudil or bimatoprost) was applied in the right eye of each participant, and the left eye served as control. At 15 and 120 min after a single instillation of the eyedrop in the right eye, IOP measurement, AS-OCTA examination, and slit-lamp photography were performed. All IOP measurements were taken with the subjects in the seated position just before slit-lamp photography and AS-OCTA examination.

### 2.3. Anterior Segment Optical Coherence Tomography Angiography (AS-OCTA) Examination

The OCTA examination was performed using a swept-source OCT system (PLEX Elite 9000; Carl Zeiss Meditec) [12,13,15,16]. This instrument has a central wavelength between 1040 and 1060 nm, a bandwidth of 100 nm, an A-scan depth of 3.0 mm in tissue, and a full-width at half-maximum axial resolution of approximately 5 μm in tissue. The instrument performs 100,000 A-scans/s. The AS-OCTA images were acquired using the 10-diopter optical adaptor lens developed by Carl Zeiss Meditec.

### 2.4. OCTA Image Acquisition and Processing

For each participant, a 3 × 3-mm scan pattern was used to acquire AS-OCTA images of the corneal limbus in the temporal and nasal regions, which consisted of 300 A-scans per B-scan repeated four times at each of the 300 B-scan positions. The lateral resolution of the image was estimated to be approximately 20 μm, whereas the axial resolution could be defined as 5 µm/pixel.

En face images were generated using built-in software (ver. 1.6 for the ripasudil instillation study and ver. 1.7 for the bimatoprost instillation study; Carl Zeiss Meditec). Flattening was performed at the level of the conjunctival epithelium, which was misidentified as the inner limiting membrane by the software. Superficial- and deep-layer flow images were developed with en face maximum projection from the conjunctival epithelium to a depth of 200 µm (mainly conjunctival composition) and from a depth of 200 µm to 1000 µm from the conjunctival epithelium (mainly intrascleral composition), respectively, as previously reported [12,13]. The projection artifact removal algorithm in the built-in software was used when developing the en face images [17].

### 2.5. Quantitative Measurements

The VDs of superficial- and deep-layer flows were measured in the 3 × 3-mm scan images with a 1024 × 1024-pixel rectangular box in the nasal and temporal quadrants (Appendix A). Each AS-OCTA image in both temporal and nasal regions was analyzed. VD was defined as the ratio of the area occupied by the vessels divided by the total area after binarization of images [18]. For binarization of images, a Trainable Weka [19], which used machine-learning algorithms, was applied in ImageJ software (Wayne Rasband, National Institutes of Health, Bethesda, MD) [20] to extract true vessels for minimizing the impact of image noise [21]. Briefly, after training for vessels and background on one selected AS-OCTA image, the classifier and data were saved on the computer. Then, the same trained classifier was applied to all AS-OCTA images (Appendix A). The classifier and data used in the current study are available at https://www.mdpi.com/. A previous study showed that the intraclass correlation coefficient (ICC) (95% confidence interval) for VD for two AS-OCTA scans obtained on the same day was 0.834 (0.708–0.908) in the superficial layer and 0.935 (0.882–0.965) in the deep layer [13]. These ICC values indicated the excellent reproducibility of AS-OCTA VDs.

### 2.6. Statistical Analysis

Differences in VD at 15 min or 2 h after instillation from the baseline were evaluated by the Wilcoxon signed-rank test. Stepwise multiple linear regression analyses were performed to identify the effects of factors with *p* < 0.1 in the univariate analyses on percent changes in IOP at 15 min and 2 h from the baseline in all 20 eyes of 10 participants in each group. The use of eye drop was added as a variable to multivariate analyses, regardless of its *p* value. All analyses were performed using IBM SPSS Statistics 24 (IBM Corp., Armonk, NY, USA). *p* values less than 0.05 were considered statistically significant, but in the post-hoc analyses in VD, *p* values less than 0.0125 were considered statistically significant after Bonferroni correction. Except where stated otherwise, data with normal distributions were presented as means (standard deviation (SD)) and those with non-parametric distributions were presented as medians (25-th percentile, 75-th percentile). Data analysis was conducted from 1 July to 28 August 2020.

## 3. Results

### 3.1. Participants and Intraocular Pressure (IOP) Measurements

Baseline characteristics at study enrolment and the time course of IOP changes are presented in Table 1. Mean IOPs (SD) in the right eye after ripasudil instillation were 13.9 (2.9) mm Hg at baseline, 12.3 (2.6) mm Hg at 15 min, and 11.3 (3.1) mm Hg at 2 h. The corresponding values after bimatoprost instillation were 12.8 (2.2) mm Hg at baseline, 11.9 (1.7) mm Hg at 15 min, and 11.7 (0.9) mm Hg at 2 h.

### 3.2. Time Course of Vasculature Findings in the Superficial and Deep Layers

Table 2 shows the time course of deep and superficial VDs at baseline, and 15 min and 2 h after ripasudil or bimatoprost instillation. In the ripasudil group, the deep VDs (median (25-th percentile, 75-th percentile)) at 15 min (19.91% (14.82%, 24.07%)) and 2 h (14.80% (12.03%, 19.98%)) after ripasudil instillation were significantly higher than that at baseline (13.11% (11.52%, 15.92%)) (*p* < 0.001 for both comparisons). The superficial VD at 15 min (37.29% (33.20%, 40.90%)) after ripasudil instillation was significantly higher (*p* < 0.001) than that at baseline (28.46% (25.08%, 35.43%)), but the superficial VD at 2 h (31.64% (27.42%, 36.75%)) was not significantly higher than that at baseline (*p* = 0.015) after Bonferroni correction.

In the bimatoprost group, compared to the deep VD (median (25th percentile, 75th percentile)) at baseline (24.21% (16.96%, 27.43%)), the deep VDs at 15 min (19.34% (15.19%, 25.06%)) and 2 h (22.07% (17.50%, 26.16%)) were slightly lower, but the differences were not statistically significant after Bonferroni correction (*p* = 0.037 and 0.48) (Table 2). The superficial VD was significantly lower at 15 min (39.53% (35.23%, 47.19%), *p* = 0.010) and higher at 2 h after bimatoprost instillation (48.19% (45.02%, 53.95%), *p* = 0.011) than the superficial VD at baseline (42.64% (39.49%, 46.84%)).

Example images from the ripasudil and bimatoprost groups are shown in Figure 1 and Figure 2, respectively. The superficial AO-OCTA images show much more vasculature than the slit-lamp photographs at each time point. After ripasudil instillation, both the superficial and deep vasculatures markedly increased at 15 min and decreased again at 2 h, but the deep vasculature was still more noticeable even at 2 h compared to the baseline (Figure 1). After bimatoprost instillation, the superficial vasculature was more noticeable at 2 h, but was almost the same level or less than baseline at 15 min (Figure 2). The deep vasculature did not change apparently over time.

### 3.3. Association between IOP Reduction and AS-OCTA Parameters

Factors associated with the percent changes in IOP after eyedrop instillations were examined (Table 3). In the ripasudil group, stepwise multiple regression analyses showed that the use of eyedrops and lower baseline deep VD were significantly associated with a greater IOP reduction at 15 min (*p* < 0.001 and 0.004) and 2 h (*p* < 0.001 and 0.018) after ripasudil instillation. The changes in deep and superficial VD at 15 min were significantly associated with the percent change in IOP only in the univariate analyses (Table 3). In the bimatoprost group, the greater IOP reduction was significantly associated with the greater baseline IOP (at 15 min, *p* < 0.001; 2 h, *p* < 0.001) and the use of eyedrops (at 2 h, *p* = 0.004) after bimatoprost instillation.

## 4. Discussion

The current study used AS-OCTA with a short observation period in normal participants and showed that ripasudil instillation increased both conjunctival and intrascleral vasculatures, whereas bimatoprost instillation affected mainly the conjunctival vasculature. The increase in deep vasculature remained at a high level even 2 h after ripasudil instillation, and the effect of ripasudil on IOP reduction was significantly associated with baseline VD in the deep layer. On the other hand, the effect of bimatoprost on IOP was not associated with any of the OCTA parameters examined. These results suggest that the IOP reduction induced by ripasudil might be closely associated with the vasculature in the deep layer.

Evaluation of conjunctival hyperemia is usually performed by a grading scale from 0 to 3, 4, or 5 units [2,3,4]; however, the interobserver correlation in the conjunctival hyperemia score was reported to be relatively low because of the subjective nature of these assessments [5]. Therefore, VD measurement using AS-OCTA shows better objectivity. Notably, the visualization of the vasculature differed between AS-OCTA and slit-lamp photographs, with superficial AS-OCTA images showing much more vasculature than slit-lamp photographs in a reproducible fashion (Figure 1 and Figure 2). This suggests that evaluation using slit-lamp photography is limited to large remarkable vessels and that this technique may miss many tiny vessels. The clinical usefulness of AS-OCTA in evaluating conjunctival hyperemia remains to be examined, and the deep vasculature, which is difficult to separately visualize through photography, may be a good target for AS-OCTA.

In this study, the deep VD as well as the superficial VD significantly increased at 15 min and remained high even 2 h after ripasudil instillation. This was consistent with the findings of a previous report, in which the ROCK inhibitor could induce dilation of ESVs in enucleated human eyes [22]. It is unclear whether this dilation of ESVs contributes to the IOP reduction. There are numerous arterio-venous anastomoses (AVAs) in the episcleral vasculature that control the flow of blood between arterioles and venules [23]. Although it has been suggested that general vasodilation of the episcleral vasculature, including the AVAs, causes an increase in both episcleral venous pressure (EVP) and IOP, it also has been hypothesized that AVA closure, which reduces blood flow from the arteriole to the venule side, could cause a reduction in EVP and IOP and that ROCK inhibitor-induced vasodilation of the episcleral vasculatures might reduce the resistance of AHO and IOP [22,24,25]. Further studies are needed to clarify the association between ESV dilation and IOP.

A lower deep VD at baseline was significantly associated with higher IOP reduction after ripasudil instillation. The reason for this association is, however, unclear. Episcleral vasculature is thought to be a key determinant of IOP, although its role in the response to ocular hypotensive therapies is not well understood [25]. Our previous report showed that a higher deep VD was significantly associated with higher IOP in treated eyes with glaucoma [13]. Since OCTA signals are mainly derived from flowing red blood cells [26], the fewer AS-OCTA-positive vessels in the deep layer might be associated with better function of the post-TM AHO. Our results showed that the vasculature in the deep layer remained at a high level even 2 h after instillation, but that in the superficial layer did not. The changes in deep VD were significantly associated with the IOP reduction in univariate analyses, but not in multivariate analysis. Therefore, we could not determine whether the increase in the deep vasculature induced by ripasudil affects IOP reduction. Nevertheless, AS-OCTA may be useful to predict the IOP-lowering effects of ripasudil eyedrops before use in the future.

After bimatoprost instillation, the superficial VD significantly increased at 2 h, which is consistent with the findings of previous studies evaluating conjunctival hyperemia using a grading scale [2,10]. On the other hand, the deep VD did not differ significantly over time. It is suggested that the conjunctival hyperemia induced by PGAs is caused by vasodilation mediated by endothelial cell-derived nitric oxide [27]. Although PGAs induced IOP reduction mainly thorough uveoscleral outflow [28], they also possibly lowered IOP through conventional AHO [29]. Thus, there might be scope for examination of the associations between deep vasculature and IOP reduction, as well as the transient reduction of superficial VD detected immediately after bimatoprost instillation in this study.

This study had several limitations. First, it included only young, healthy participants and used a small sample size. To reveal the usefulness of AS-OCTA in clinical practice for glaucoma, further large-scale studies including glaucoma patients are needed. Second, because of the absence of commercially available specialized software for AS-OCTA, we used software dedicated to the posterior segment. We distinguished between the superficial and deep layers on the basis of the depth from the conjunctival epithelium and, therefore, could not consider the possible changes in the conjunctival thickness. Future improvement of the algorithm specialized for the AS might make AS-OCTA more useful. Third, the ripasudil instillation study and bimatoprost instillation study were performed at different times, and a minor upgrade of the software version (from ver. 1.6 to 1.7) and fine adjustment of the laser power was performed between the two studies. These factors might have been one of the reasons for the differences in deep and superficial VDs at the baseline between the groups (Table 2). However, they would not have influenced our results because the conditions were the same in each group. Nevertheless, it is worth noting that even slightly different conditions can significantly affect AS-OCTA VD measurements.

## 5. Conclusions

AS-OCTA can be used for longitudinal assessments of conjunctival and intrascleral vasculatures, and the AS-OCTA findings in this study revealed that the patterns of hyperemia caused by ripasudil and bimatoprost were different in timing, depth, and extent. Deep OCTA flow signals were closely associated with IOP changes after ripasudil instillation. Further large-scale studies are needed to confirm whether AS-OCTA can be useful to predict IOP-lowering effects of a certain type of anti-glaucoma eyedrop.

## Figures and Tables

**Figure 1 jcm-09-04016-f001:**
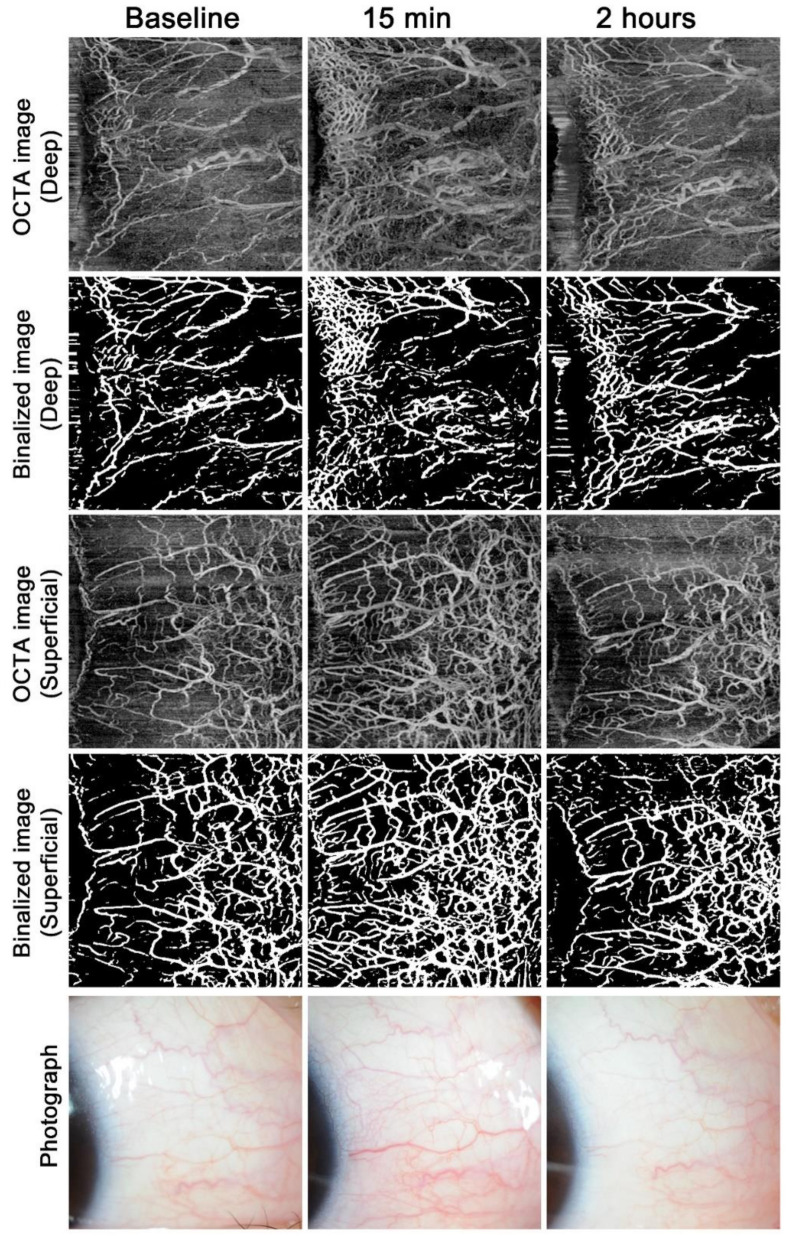
Time course of anterior-segment optical coherence tomography angiography findings and photographs after ripasudil instillation. Example images in the nasal region of the right eye in a normal participant after a single instillation of ripasudil.

**Figure 2 jcm-09-04016-f002:**
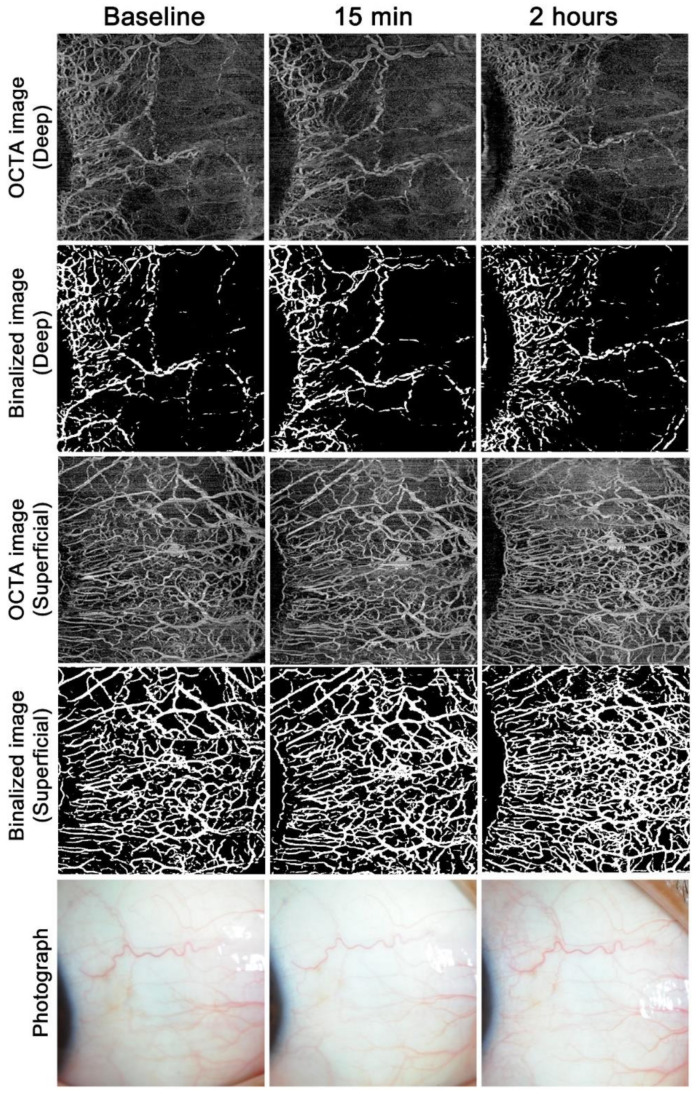
Time course of anterior-segment optical coherence tomography angiography findings and photographs after bimatoprost instillation. Example images in the nasal region of the right eye in a normal participant after a single instillation of bimatoprost.

**Table 1 jcm-09-04016-t001:** Baseline characteristics of participants and course of intraocular pressure.

	Ripasudil Group		Bimatoprost Group	
Age, mean (SD), y	30.1 (7.8)		27.6 (5.4)	
Sex, *n*, male/female	5/5		4/6	
	Right eye	Left eye	Right eye	Left eye
Axial length, mean (SD), mm	24.81 (1.24)	24.87 (1.23)	24.77 (1.04)	24.76 (0.92)
Central corneal thickness, mean (SD), μm	550.2 (32.3)	551.2 (29.5)	547.4 (31.2)	546.6 (31.0)
Baseline IOP, mean (SD), mm Hg	13.9 (2.9)	14.0 (2.5)	12.8 (2.2)	13.0 (2.4)
IOP 15 min, mean (SD), mm Hg	12.3 (2.6)	13.6 (2.3)	11.9 (1.7)	12.6 (1.2)
IOP 2 h, mean (SD), mm Hg	11.3 (3.1)	13.2 (2.6)	11.7 (0.9)	12.6 (1.5)

Abbreviations: IOP = intraocular pressure; SD = standard deviation.

**Table 2 jcm-09-04016-t002:** Vessel densities (VDs) in the deep and superficial layers after ripasudil or bimatoprost instillation.

Eye Drop and Side	Median (25th Percentile, 75th Percentile)			*p* Value ^ab^	*p* Value ^ac^
	At Baseline	At 15 min	At 2 h		
Ripasudil group Right eye					
Deep VD, %	13.11% (11.52%, 15.92%)	19.91% (14.82%, 24.07%)	14.80% (12.03%, 19.98%)	**<0.001**	**0.009**
Superficial VD, %	28.46% (25.08%, 35.43%)	37.29% (33.2%, 40.90%)	31.64% (27.42%, 36.75%)	**<0.001**	0.015
Ripasudil group Left eye					
Deep VD, %	14.28% (11.83%, 16.45%)	13.49% (10.61%, 18.16%)	13.52% (10.41%, 16.94%)	0.33	0.85
Superficial VD, %	29.45% (25.04%, 33.64%)	27.27% (23.16%, 33.69%)	29.56% (26.78%, 33.90%)	0.093	0.91
Bimatoprost group Right eye					
Deep VD, %	24.21% (16.96%, 27.43%)	19.34% (15.19%, 25.06%)	22.07% (17.50%, 26.16%)	0.037	0.48
Superficial VD, %	42.64% (39.49%, 46.84%)	39.53% (35.23%, 47.19%)	48.19% (45.02%, 53.95%)	**0.010**	**0.011**
Bimatoprost group Left eye					
Deep VD, %	20.55% (17.01%, 25.87%)	23.41% (16.24%, 31.80%)	22.40% (18.68%, 30.80%)	0.18	0.10
Superficial VD, %	42.73% (38.25%, 46.25%)	42.28% (37.09%, 48.51%)	46.46% (39.54%, 49.09%)	0.85	0.067

^a^ Calculated using the Wilcoxon signed-rank test. Values that are statistically significant are shown in bold, after setting the limit for false-positive error to 0.0125 after Bonferroni correction for multiple analyses. ^b^ Comparison between findings at baseline and at 15 min. ^c^ Comparison between findings at baseline and at 2 h.

**Table 3 jcm-09-04016-t003:** Factors associated with the percent changes in intraocular pressure (IOP).

Ripasudil	Percent IOP Change at 15 min						Percent IOP Change at 2 h					
	Univariate Analysis			Multivariable Analysis ^a^			Univariate Analysis			Multivariable Analysis ^a^		
	B	β	*p*	B	β	*p*	B	β	*p*	B	β	*p*
Use of eye drop	**−8.677**	**−0.657**	**<0.001**	**−8.191**	**−0.620**	**<0.001**	**−13.034**	**−0.521**	**0.001**	**−11.822**	**−0.473**	**0.001**
Baseline IOP	−0.419	−0.163	0.32	-	-	-	−0.018	−0.004	0.98	-	-	-
Baseline deep VD	**0.366**	**0.34**	**0.032**	**0.267**	**0.248**	**0.044**	**0.808**	**0.396**	**0.011**	**0.665**	**0.326**	**0.018**
Baseline superficial VD	0.064	0.077	0.64	-	-	-	0.486	0.308	0.053			0.72
Change in deep VD at 15 min	**−0.031**	**−0.36**	**0.023**			0.70	−0.025	−0.155	0.34	-	-	-
Change in deep VD at 2 h	−0.014	−0.086	0.60	-	-	-	0.033	0.106	0.51	-	-	-
Change in superficial VD at 15 min	**−0.094**	**−0.468**	**0.002**			0.61	**−0.151**	**−0.398**	**0.011**			0.61
Change in superficial VD at 2 h	−0.053	−0.236	0.14	-	-	-	−0.109	−0.254	0.11	-	-	-
**Bimatoprost**	**Percent IOP Change at 15 min**						**Percent IOP Change at 2 h**					
	**Univariate Analysis**			**Multivariable Analysis ^a^**			**Univariate Analysis**			**Multivariable Analysis ^a^**		
	**B**	**β**	***p***	**B**	**β**	***p***	**B**	**β**	***p***	**B**	**β**	***p***
Use of eye drop	−4.274	−0.092	0.57			0.070	−6.715	−0.12	**0.46**	**−9.587**	**−** **0.171**	**0.004**
Baseline IOP	**−9.363**	**−0.878**	**<** **0** **.001**	**−9.363**	**−0.878**	**<0.001**	**−11.844**	**−0.927**	**<0.001**	**−11.964**	**−0.936**	**<** **0** **.001**
Baseline deep VD	−1.03	−0.304	0.056			0.74	**−1.416**	**−0.349**	**0.027**			0.89
Baseline superficial VD	0.167	0.054	0.74	-	-	-	0.628	0.170	0.29	-	-	-
Change in deep VD at 15 min	0.014	0.024	0.89	-	-	-	−0.013	−0.019	0.91	-	-	-
Change in deep VD at 2 h	0.063	0.086	0.60	-	-	-	0.054	0.062	0.70	-	-	-
Change in superficial VD at 15 min	0.265	0.194	0.23	-	-	-	0.283	0.173	0.29	-	-	-
Change in superficial VD at 2 h	−0.185	−0.110	0.50	-	-	-	−0.187	−0.093	0.57	-	-	-

Abbreviations: VD = vessel density; Β = unstandardized regression coefficient; β = standardized regression coefficient. Values that are statistically significant are shown in bold. ^a^ Stepwise regression analysis for all variables with *p* < 0.1 in a univariable regression model with use of eyedrops.

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
