# Peer review of "Short-Term Effects of Different Types of Anti-Glaucoma Eyedrop on the Sclero-Conjunctival Vasculature Assessed Using Anterior Segment OCTA in Normal Human Eyes: A Pilot Study"

_jcm, 2020, doi:10.3390/jcm9124016_

Round 1
Reviewer 1 Report
This appears to be a nice pilot study indicating proof-of-concept that OCT-A may be useful to predict IOP response for ROCK inhibitors.
I believe that if validated in actual patients with glaucoma and with a much larger number of eyes studied, this may have potential for impact in real-world clinics.
I would encourage the authors to pursue a larger clinical trial in subjects with diagnosed glaucoma.
Author Response
Reviewer 1
This appears to be a nice pilot study indicating proof-of-concept that OCT-A may be useful to predict IOP response for ROCK inhibitors.
I believe that if validated in actual patients with glaucoma and with a much larger number of eyes studied, this may have potential for impact in real-world clinics.
I would encourage the authors to pursue a larger clinical trial in subjects with diagnosed glaucoma.
> We thank the reviewer for the supportive comments. We will plan a larger clinical trial in the near future.
Reviewer 2 Report
The authors prospectively investigated the time-dependent changes in superficial and deep AS-OCTA flow images and the associations between changes in IOP and vessel density (VD) by using a swept-source optical coherence tomography (OCT) system after a single instillation of ripasudil or bimatoprost in healthy subject. As a result, the superficial and deep VD responded differently to the two drugs. Deep VD were closely associated with IOP changes after ripasudil installation.
They showed usefulness of AS-OCTA for longitudinal assessments of conjunctival and intrascleral vasculatures. This paper and their previous works in AS-OCT are novel in its approach.
Major issue
- At baseline VD measurment, the two drug groups had different superficial and deep VDs. We cannot evaluate whether there is a selection bias, even if the comparison target is the opposite eye. The group difference could be statistically evaluated. Authors also could show or compare a kind of VD normative distribution.
- Explanation of IOP measurement procedure is needed in method section. Ophthalmologists know the ambiguity of IOP measurement, but others not. Why did authors choose Icare pro rebound tonometer instead of goldmann applanation tonometer?
Minor issue
- I am concerned about tissue thickness change. After drug installation, thickness of conjunctiva might increase with increased aqueous humor outflow. Is there any data or discussion?
Author Response
Reviewer 2
The authors prospectively investigated the time-dependent changes in superficial and deep AS-OCTA flow images and the associations between changes in IOP and vessel density (VD) by using a swept-source optical coherence tomography (OCT) system after a single instillation of ripasudil or bimatoprost in healthy subject. As a result, the superficial and deep VD responded differently to the two drugs. Deep VD were closely associated with IOP changes after ripasudil installation.
They showed usefulness of AS-OCTA for longitudinal assessments of conjunctival and intrascleral vasculatures. This paper and their previous works in AS-OCT are novel in its approach.
> We thank the reviewer for the supportive comments.
Major issue
At baseline VD measurment, the two drug groups had different superficial and deep VDs. We cannot evaluate whether there is a selection bias, even if the comparison target is the opposite eye. The group difference could be statistically evaluated. Authors also could show or compare a kind of VD normative distribution.
> We thank the reviewer for pointing out this important issue. The ripasudil instillation study and the bimatoprost instillation study were performed at different times, as shown in lines 65-66:
“The ripasudil instillation study was performed between October 1, 2018, and January 30, 2019, and the bimatoprost instillation study was performed between December 1, 2019, and February 28, 2020.”
We found that a minor version upgrade (ver. 1.6 to 1.7) and fine adjustment of the laser power had been performed between these two instillation studies. These differences might be one of the reasons for the differences in baseline deep and superficial VDs between the groups. We believe that these differences would not influence our conclusions since the conditions were the same within each group. Nevertheless, we have added this information as a limitation in lines 246-251.
Explanation of IOP measurement procedure is needed in method section. Ophthalmologists know the ambiguity of IOP measurement, but others not. Why did authors choose Icare pro rebound tonometer instead of goldmann applanation tonometer?
> We thank the reviewer for this important suggestion. The Icare PRO Rebound Tonometer offers an advantage of obtaining measurements without changing the participant’s seat and with minimal effects on the ocular surface. We have added the description about choosing it for IOP measurement in lines 71-80.
Minor issue
I am concerned about tissue thickness change. After drug installation, thickness of conjunctiva might increase with increased aqueous humor outflow. Is there any data or discussion?
> Thank you for pointing this out. We did not have data about the thickness of the conjunctiva. We have added this issue as a limitation in lines 243-245.
Reviewer 3 Report
Comments to Authors
This pilot study explores the possibility of using anterior segment OCTA (AS-OCTA) as a clinical tool to predict the IOP-lowering effects of pressure-reducing drugs before use. Such a capability has clinical relevance because, if achievable, it would spare patients from starting an ineffective medication and also eliminate the need for return visits to determine drug effectiveness.
AS-OCTA was used to determine vessel density of the superficial and deep vasculature of the distal outflow pathways after instillation of ripasudil or bimatoprost and then explore correlations with changes in IOP. The authors conclude that AS-OCTA can detect differences in timing, depth, and extent of effects.
The authors appropriately call attention to limitations, including a small sample size, young healthy participates, and the lack of inclusion of glaucoma patients. These are all issues that can be remedied in future studies. The lack of dedicated software is an additional limitation that could be remedied in the future, especially if additional studies provide evidence of the technique's potential clinical usage.
Comments:
Line 14 This appears to be a short-term experimental study of the acute effects of medications rather than what is more typically thought of as a longitudinal study. Would it be appropriate to leave out the word "longitudinal here"?
Line 52 Li et al. were the first to study the feasibility of microvascular imaging in human corneoscleral limbus using OCTA. The study needs to be recognized in the paper. P Li, L An, R Reif, TT Shen, M Johnstone, RK Wang. "In vivo microstructural and microvascular imaging of the human corneoscleral limbus using optical coherence tomography." Biomedical optics express 2 (11), 3109-3118 (2011)
Line 85-86: "The AS-OCTA images were acquired using the 10-diopter optical adaptor lens developed by Carl Zeiss 85 Meditec.". The lateral resolution provided by this attachment should be provided. This evaluation can be done by a simple measurement of a knife-edge. The lateral resolution is important in your context because vessel area density is assessed in your investigation.
Line 97-98: "The projection-resolved algorithm in the built-in software was used when developing the en face images." "projection-resolved algorithm" should be called "projection artifact removal algorithm." This paper below should be cited to help readers to follow this algorithm. A Zhang, Q Zhang, RK Wang. "Minimizing projection artifacts for accurate presentation of choroidal neovascularization in OCT micro-angiography." Biomedical optics express 6 (10), 4130-4143 (2015)
Section 2.5: When performing quantification of vascular features in OCTA images from Zeiss machines, the below paper should be cited to help readers to follow: Z Chu, et al. "Quantitative assessment of the retinal microvasculature using optical coherence tomography angiography." Journal of biomedical optics 21 (6), 066008 (2016).
VAD is a useful metric in assessing the effect of anti-glaucoma eyedrops on the microvasculature within the corneoscleral limbus. To make your assessment more credible, it is strongly suggested to conduct a repeatability and reproducibility test for the VAD measurements in both superficial and deep vascular layers.
Author Response
Reviewer 3
This pilot study explores the possibility of using anterior segment OCTA (AS-OCTA) as a clinical tool to predict the IOP-lowering effects of pressure-reducing drugs before use. Such a capability has clinical relevance because, if achievable, it would spare patients from starting an ineffective medication and also eliminate the need for return visits to determine drug effectiveness.
AS-OCTA was used to determine vessel density of the superficial and deep vasculature of the distal outflow pathways after instillation of ripasudil or bimatoprost and then explore correlations with changes in IOP. The authors conclude that AS-OCTA can detect differences in timing, depth, and extent of effects.
The authors appropriately call attention to limitations, including a small sample size, young healthy participates, and the lack of inclusion of glaucoma patients. These are all issues that can be remedied in future studies. The lack of dedicated software is an additional limitation that could be remedied in the future, especially if additional studies provide evidence of the technique's potential clinical usage.
> We thank the reviewer for the positive comments.
Comments:
Line 14 This appears to be a short-term experimental study of the acute effects of medications rather than what is more typically thought of as a longitudinal study. Would it be appropriate to leave out the word "longitudinal here"?
> We have removed “longitudinal,” as suggested.
Line 52 Li et al. were the first to study the feasibility of microvascular imaging in human corneoscleral limbus using OCTA. The study needs to be recognized in the paper. P Li, L An, R Reif, TT Shen, M Johnstone, RK Wang. "In vivo microstructural and microvascular imaging of the human corneoscleral limbus using optical coherence tomography." Biomedical optics express 2 (11), 3109-3118 (2011)
> We thank the reviewer for this suggestion. We have revised the manuscript and added the suggested paper as reference 11.
Line 85-86: "The AS-OCTA images were acquired using the 10-diopter optical adaptor lens developed by Carl Zeiss 85 Meditec.". The lateral resolution provided by this attachment should be provided. This evaluation can be done by a simple measurement of a knife-edge. The lateral resolution is important in your context because vessel area density is assessed in your investigation.
> We have added the description in line 87 as follows:
“and the lateral resolution of the image was estimated to be approximately 20 μm”
Line 97-98: "The projection-resolved algorithm in the built-in software was used when developing the en face images." "projection-resolved algorithm" should be called "projection artifact removal algorithm." This paper below should be cited to help readers to follow this algorithm. A Zhang, Q Zhang, RK Wang. "Minimizing projection artifacts for accurate presentation of choroidal neovascularization in OCT micro-angiography." Biomedical optics express 6 (10), 4130-4143 (2015)
> We have changed “projection-resolved algorithm” to “projection artifact removal algorithm” (line 99) and we have added the suggested paper as reference 17.
Section 2.5: When performing quantification of vascular features in OCTA images from Zeiss machines, the below paper should be cited to help readers to follow: Z Chu, et al. "Quantitative assessment of the retinal microvasculature using optical coherence tomography angiography." Journal of biomedical optics 21 (6), 066008 (2016).
> We have added the suggested paper as reference 18.
VAD is a useful metric in assessing the effect of anti-glaucoma eyedrops on the microvasculature within the corneoscleral limbus. To make your assessment more credible, it is strongly suggested to conduct a repeatability and reproducibility test for the VAD measurements in both superficial and deep vascular layers.
> We agree with this suggestion. We had previously reported intraclass correlation coefficient (ICC) values to evaluate the reproducibility of AS-OCTA VD measurements. We have added the following description in lines 113-116.
“A previous study showed that the intraclass correlation coefficient (ICC) (95% confidence interval) for VD for two AS-OCTA scans obtained on the same day was 0.834 (0.708–0.908) in the superficial layer and 0.935 (0.882–0.965) in the deep layer [13]. These ICC values indicated the excellent reproducibility of AS-OCTA VDs.”
Reviewer 4 Report
interesting work however there are somethings to think of.
there are only 5 in each group it is too small for the conclusion. as to the small sample size even the baseline is quite different with each group. authors need to add more patients for proper conclusion.
what is the clinical value of this study ? as author said that "Since deep OCTA flow signals are closely associated with IOP changes after ripasudil instillation, AS-OCTA may be useful to predict its IOP-lowering effects"
do you really think you can predict the IOP reduction rate with AS-OCTA? or do you really think all low deep VD will result in hier IOP lowering effect? if you think so, what is the reason for this?
Author Response
Reviewer 4
interesting work however there are somethings to think of.
there are only 5 in each group it is too small for the conclusion. as to the small sample size even the baseline is quite different with each group. authors need to add more patients for proper conclusion.
> We included 10 patients in each study group as described in our manuscript. We agree that the sample size was very small since this was a pilot study; this has been described as a limitation in the manuscript. The VDs at the baseline were indeed different between the groups. This issue has also been pointed out by Reviewer 2, and we have provided the following response.
“The ripasudil instillation study and the bimatoprost instillation study were performed at different times as shown in lines 65-66:
‘The ripasudil instillation study was performed between October 1, 2018, and January 30, 2019, and the bimatoprost instillation study was performed between December 1, 2019, and February 28, 2020.’
We found that a minor version upgrade (ver. 1.6 to 1.7) and fine adjustment of the laser power had been performed between these two instillation studies. These differences might be one of the reasons for the differences in baseline deep and superficial VDs between the groups. We believe that these differences would not influence our conclusions since the conditions were the same within each group. Nevertheless, we have added this information as a limitation in lines 246-251.”
what is the clinical value of this study ? as author said that "Since deep OCTA flow signals are closely associated with IOP changes after ripasudil instillation, AS-OCTA may be useful to predict its IOP-lowering effects"
> Reviewer 3 kindly summarized the study with the comment, “This pilot study explores the possibility of using anterior segment OCTA (AS-OCTA) as a clinical tool to predict the IOP-lowering effects of pressure-reducing drugs before use. Such a capability has clinical relevance because, if achievable, it would spare patients from starting an ineffective medication and also eliminate the need for return visits to determine drug effectiveness.” We think that this comment adequately explains the clinical value of this study.
do you really think you can predict the IOP reduction rate with AS-OCTA? or do you really think all low deep VD will result in hier IOP lowering effect? if you think so, what is the reason for this?
> Since this was a pilot study, we do not have enough data to answer this question. However, considering the fact that IOP is affected by the function of the post-TM AHO, and AS-OCTA images are closely related to the post-TM AHO, we strongly believe that AS-OCTA can be helpful in predicting the IOP-lowering effect of eye drops that act on the conventional AHO.
Round 2
Reviewer 2 Report
The manuscript has been revised well. I think this manuscript will be acceptable after some corrections have been done.
Author Response
Reviewer 2
The manuscript has been revised well. I think this manuscript will be acceptable after some corrections have been done.
> We thank the reviewer for the favorable comment. Because Reviewer 4 commented that more patients are needed for our conclusions, we have revised the conclusions as follows:
In the previous version, “Since deep OCTA flow signals are closely associated with IOP changes after ripasudil instillation, AS-OCTA may be useful to predict its IOP-lowering effects.”
In the revised version, “Deep OCTA flow signals were closely associated with IOP changes after ripasudil instillation. Further large-scale studies are needed to confirm whether AS-OCTA can be useful to predict IOP-lowering effects of a certain type of anti-glaucoma eyedrop.”
Reviewer 4 Report
i still think more patients are needed but however the study is interesting
Author Response
Reviewer 4
i still think more patients are needed but however the study is interesting
> We thank the reviewer for the suggestion. We have revised the conclusion as follows:
In the previous version, “Since deep OCTA flow signals are closely associated with IOP changes after ripasudil instillation, AS-OCTA may be useful to predict its IOP-lowering effects.”
In the revised version, “Deep OCTA flow signals were closely associated with IOP changes after ripasudil instillation. Further large-scale studies are needed to confirm whether AS-OCTA can be useful to predict IOP-lowering effects of a certain type of anti-glaucoma eyedrop.”